# Primary care system-level training and support programme for the secondary prevention of domestic violence and abuse: a cost-effectiveness feasibility model

Madeleine Cochrane ![ORCID],[1] Eszter Szilassy,[1] Caroline Coope ![ORCID],[1] Elizabeth Emsley,[1] Medina Johnson,[2] Gene Feder ![ORCID],[1] Estela Capelas Barbosa[1]

¹Population health Sciences, University of Bristol, Bristol, UK
²IRISi, London, UK

**Correspondence to**
Dr Madeleine Cochrane;
madeleine.cochrane@bristol.ac.uk

## ABSTRACT

**Objectives** This study aimed to evaluate the prospective cost-effectiveness of the Identification and Referral to Improve Safety plus (IRIS+) intervention compared with usual care using feasibility data derived from seven UK general practice sites.

**Method** A cost–utility analysis was conducted to assess the potential cost-effectiveness of IRIS+, an enhanced model of the UK's usual care. IRIS+ assisted primary care staff in identifying, documenting and referring not only women, but also men and children who may have experienced domestic violence/abuse as victims, perpetrators or both. A perpetrator group programme was not part of the intervention per se but was linked to the IRIS+ intervention via a referral pathway and signposting. A Markov model was constructed from a societal perspective to estimate mean incremental costs and quality-adjusted life years (QALYs) of IRIS+ compared with to usual care over a 10-year time horizon.

**Results** The IRIS+ intervention saved £92 per patient and produced QALY gains of 0.003. The incremental net monetary benefit was positive (£145) and the IRIS+ intervention was cost-effective in 55% of simulations at a cost-effectiveness threshold of £20 000 per QALY.

**Conclusion** The IRIS+ intervention could be cost-effective or even cost saving from a societal perspective in the UK, though there are large uncertainties, reflected in the confidence intervals and simulation results.

## BACKGROUND

Domestic violence/abuse (DVA) is a public health challenge, affecting approximately 9 million adults and 2 million children in the UK.[1–4] The societal cost of DVA was estimated by the UK Home Office to be £66bn in 2017, not including costs to children. Safe Lives, a UK-wide DVA charity, highlighted the need for an initial £2.2bn of public investment per annum to cover domestic abuse services for the whole family- adult, teen and child victims, and perpetrators.[5 6] Public Health England identified primary care as a key

location for interventions to prevent DVA and improve health outcomes for adults and children. Early intervention in DVA, for example, in the primary care setting, reduces the overall public service burden of abuse and can reduce escalation of violence.[7]

## STRENGTHS AND LIMITATIONS OF THIS STUDY

⇒ To the author's knowledge, this is the first study to assess the potential cost-effectiveness of a primary care intervention providing support to all women, men and their children experiencing domestic violence/abuse.
⇒ The study draws on the structure of a previous domestic violence/abuse model, which has been published in peer-reviewed journals.
⇒ The study relies on newly collected data, reducing the need for using out-of-date previously published estimates.
⇒ The small number of newly collected data means our results may not be representative of the wider UK population.

DVA interventions to date have prioritised women, who are disproportionately affected in prevalence and severity of DVA, compared with other groups.[8 9] Identifying female survivors in primary care and referring to specialist support is effective and cost-effective through the provision of DVA training linked with a direct pathway to local DVA support.[10] The leading service model in the UK's National Health Service (NHS) primary care setting is IRIS (Identification and Referral to Improve Safety), a widely commissioned evidence-based DVA training and advocacy support programme for female survivors.

While there is a growing success in identifying women affected by DVA, male survivors and children/young people (CYP) are rarely identified in primary care and referred for

specialist support. This neglects the mental and physical health impact across the life course for CYP who experience or witness DVA[11 12] and the significant mental health impact on men exposed to DVA.[13–16] IRIS plus (IRIS+) was an enhanced model of the existing IRIS programme and was piloted in NHS primary care general practice (GP) sites, three sites in England and four sites in Wales. IRIS+ assisted GP practice staff in identifying, documenting and referring not only women, but also men and children who may have experienced DVA as victims, perpetrators or both. The IRIS+ pilot study showed feasibility and acceptability of the intervention to clinicians and those affected by DVA.[17 18] The aim of this study was to evaluate the prospective cost-effectiveness of the IRIS+ intervention when compared with usual care (the IRIS intervention). This study addresses a gap in the literature around the possible cost-effectiveness of interventions targeting men and children as well as women experiencing DVA.

## METHODS
### Overview of economic evaluation
This study was a model-based cost–utility analysis, comparing the IRIS+ intervention to usual care (the IRIS intervention). An unpublished health economic analysis plan was developed prospectively to guide the economic evaluation. The outcome measure was quality-adjusted life years (QALYs), which is the recommended outcome for economic evaluations in the UK.[19] As many of the costs of DVA are borne outside the health system, the analysis was undertaken from a UK societal perspective which in this study we define as the costs associated with implementing the intervention, downstream multisector costs associated with DVA, as well as productivity costs. Costs relating to DVA perpetration were included in the cost of onward referral, given that a perpetrator programme was linked to IRIS+ via an onward referral pathway or signposting. Costs were calculated in 2019/2020 UK£, as most of the IRIS+ intervention took place in those years. Costs and benefits were calculated over a 10-year time horizon. This was considered appropriate because the occurrence of new cases and transition probabilities were assumed to remain constant over time, and therefore, the impact of a longer time horizon would be small. While this is likely to be the case for adults, we acknowledge that the time horizon for children may be longer.[20] This means we opted for a conservative estimate of the cost-effectiveness of the intervention as far as children are concerned. Future costs and outcomes were discounted at an annual rate of 3.5% as recommended in the UK guidelines for conducting economic evaluations.[19]

### Model structure
We developed a Markov model based on the previous analysis of the cost-effectiveness of the usual care intervention (IRIS).[21 22] The model has five health states (see figure 1 for details) and the cycle length was six months, which reflects the average length of support received from

advocacy services following referral. The cycle length of six months also reflects the maximum time of support available for identified patients. Other than death, which is an absorbing state, men, women and children can transition between states in half-yearly cycles. The states were 'no abuse', 'abuse not identified', 'abuse identified and seeing advocate', 'abuse identified, not seeing advocate' and 'dead' (figure 1). A hypothetical cohort of 10 000 people was simulated moving between the states (figure 1). We used the census figures to estimate the proportion of adult men, women and children within this hypothetical cohort.[14] The model was built and run using Excel Visual Basic for Applications (VBA).

### Interventions
#### The IRIS intervention (usual care arm)
The IRIS intervention is a multicomponent intervention, which has been described elsewhere.[21 23] In short, it is delivered in UK NHS primary care GP sites and consists of multidisciplinary training sessions, targeted at the clinical team and some GP reception staff. The training sessions were designed to address barriers to improving the response of clinicians to women experiencing abuse through improved identification, support and referral to specialist agencies. Clinicians are trained to have a low threshold for asking about DVA. Training incorporates case studies and practice in asking about violence and responding appropriately. They are delivered by an advocate educator from collaborating specialist support services. The advocate educator is central to the IRIS intervention, combining a training and support role to the practices with the provision of advocacy to women referred. Ongoing support to clinicians and reception staff in the practices is provided by the advocate educator.

#### The IRIS+ intervention (intervention arm)
The IRIS+ intervention builds on the IRIS model, but in addition provides a service for men and children. Similar to IRIS, it consists of a multicomponent intervention, including multidisciplinary training for clinicians and GP staff. IRIS+ provides a simple pathway of referrals to specialist support services for women, men and their children who experience (survivors and perpetrators) DVA. In IRIS+, as well as the advocate educator, there is a dedicated children's worker. Jointly they support any referral made by clinicians, regardless of gender or age. While perpetrators could have been identified by the IRIS+ intervention, a perpetrator group programme was not part of the intervention per se but was linked to the IRIS+ intervention via a referral pathway and signposting. Perpetrators could also self-refer to the perpetrator programme.

#### Comparisons between IRIS+ (intervention arm) and IRIS (usual care arm)
Given that this study was a pilot, we did not recruit practices into the usual care arm (IRIS). In fact, the recruitment for IRIS+ included seven practices, three non-IRIS

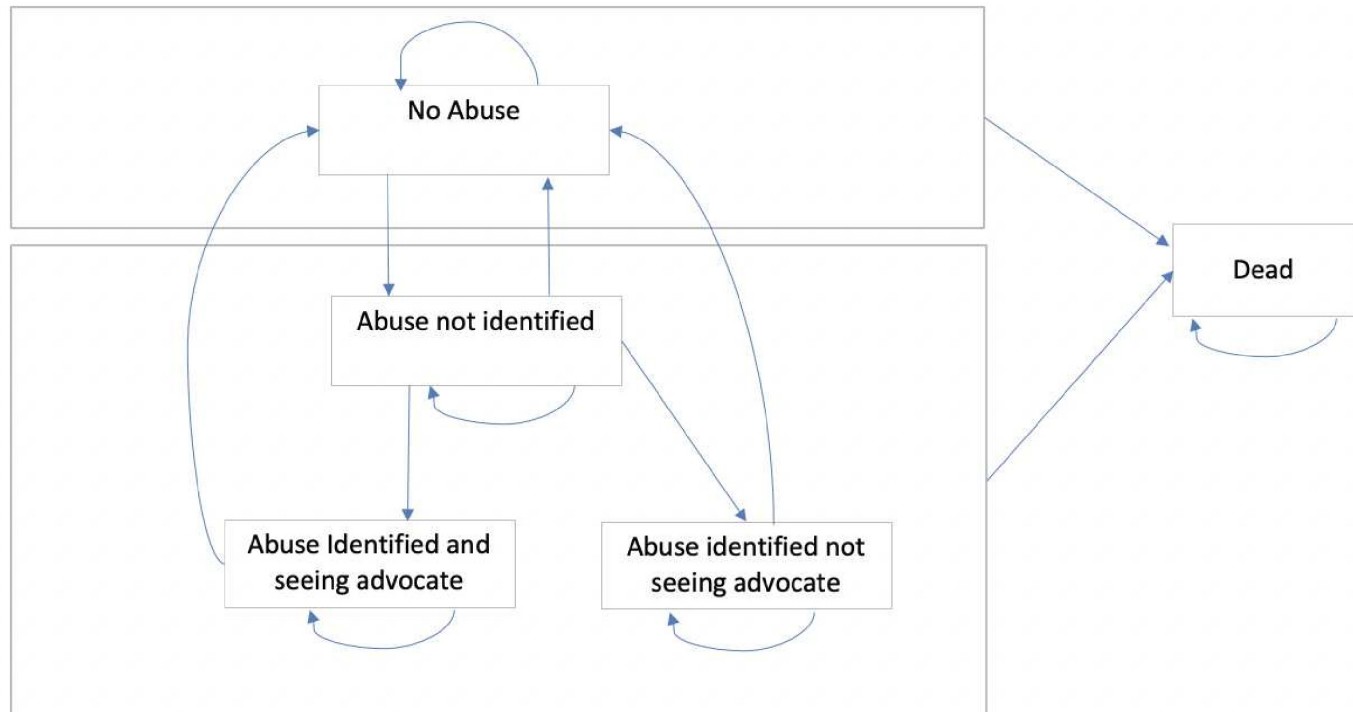

**Figure 1** Health states and movement between health states in Markov model. The model starts with all patients in either the 'no abuse' state or one of the states associated with abuse, based on the prevalence of DV (see text). Patients in the 'no abuse' state could stay in this state, move to 'abuse not identified' or die from any cause. Once a patient is in the 'abuse not unidentified' state, they could stay in that state, move back to 'no abuse', move to 'abuse identified and seeing advocate' or 'abuse identified, not seeing advocate' or die. Patients in the 'abuse identified' states could stay in these states, move back to 'no abuse' or die. Death is an absorbing state. DV, domestic violence.

trained practices that had not previously received IRIS or practice-based DVA interventions, and four IRIS-trained practices that had previously received IRIS training. The comparison between IRIS (usual care) and IRIS+ used estimated parameters based on the same areas, given both IRIS and IRIS+ programmes were available for this subset of practices.

## Parameters

Whenever possible, we used data collected in the pilot to estimate transition probabilities, utilities and costs required for the Markov model. Where this was not possible probabilities were obtained from published sources. Table 1 shows the source of data for each relevant parameter. Online supplemental tables S1 and S2 report the same parameters, however, they are reported in separate tables for adults and children, respectively.

## Prevalence of domestic abuse

The proportion of adults aged 16 years or older experiencing abuse was estimated from the Crime Survey for England and Wales (CSEW).[14] There was a subsequent published estimate, but due to anticontagion measures relating to the COVID-19 pandemic, the survey had to be moved to telephone survey, preventing some of the collection of relevant data on domestic violence. In 2018/2019, 5.5% of adults experienced some form of

domestic violence according to the CSEW.[14] Since IRIS+ also provides support services for children, we relied on the published estimate of 8% of children experiencing some form of DVA in the past 12 months.[24] Children represent 20% of the UK population.[25] To extrapolate beyond age 65, we used data from the USA showing that the prevalence of intimate partner violence was 2.2% among people aged 65 or older.[26] We, therefore, estimate that 5.5% of the UK population would be in any of the three states in the Markov model associated with abuse in the first model cycle.

## Transition probabilities

Table 1 reports all transition probabilities. There are eight transitions between states in the model, measured as follows:

1. No abuse to abuse not identified.
   No data were available to reliably estimate this probability. We, thus, estimated it using the model calibration method described below.
2. Abuse not identified to abuse identified and seeing advocate.
   For those receiving the IRIS+ intervention, we estimated this transition probability based on the number of patients seen by the advocate in IRIS+ pilot. Dividing this number by the total number of eligible patients

**Table 1** Model input parameters: probabilities

| Parameter | Base case value | Lower limit | Upper limit | Distribution | Source |
|---|---|---|---|---|---|
| Probabilities | | | | | |
| Proportion of people experiencing abuse—all ages | 0.055 | 0.038 | 0.106 | Beta | Adjusted estimate* |
| Prevalence of DVA in adults (males and females)—aged 16–90+ years | 0.055 | 0.036 | 0.073 | Beta | 14 |
| Prevalence of children exposed to DVA | 0.080 | 0.040 | 0.140 | Beta | 24 |
| Starting distribution for those experiencing abuse—all ages | | | | | |
| Abused and identified, seeing advocate | 0.003 | 0 | 0.0066 | Uniform | † |
| Abused and identified, not seeing advocate | 0.033 | 0 | 0.0660 | Uniform | † |
| Abused but not identified | 0.964 | – | – | Uniform | Complement |
| Transition probabilities—intervention and control | | | | | |
| Not abused to abused but not identified | 0.0037 | 0.0004 | 0.0106 | Dirichlet | † |
| Not abused to dead | 0.0052 | 0.0027 | 0.0087 | Dirichlet | 14 |
| Stay in not abused | 0.9911 | – | – | Dirichlet | Complement |
| Abused and identified, seeing advocate to not abused | 0.1408 | 0.0707 | 0.2301 | Dirichlet | 27 |
| Abused and identified, seeing advocate to dead | 0.0052 | 0.0000 | 0.0299 | Dirichlet | 14 |
| Stay in Abused and identified, seeing advocate | 0.8540 | – | – | Dirichlet | Complement |
| Abused and identified, not seeing advocate to not abused | 0.0781 | 0.0136 | 0.1912 | Dirichlet | 27 |
| Abused and identified, not seeing advocate to dead | 0.0052 | 0.0000 | 0.0424 | Dirichlet | 14 |
| Stay in abused and identified, not seeing advocate | 0.9167 | – | – | Dirichlet | Complement |
| Transition probabilities—intervention | | | | | |
| Abused but not identified to not abused | 0.0500 | 0.0412 | 0.0596 | Dirichlet | † |
| Abused but not identified to abused and identified, not seeing advocate | 0.0374 | 0.0298 | 0.0458 | Dirichlet | IRIS+ data |
| Abused but not identified to abused and identified, seeing advocate | 0.0312 | 0.0243 | 0.0390 | Dirichlet | IRIS+ data |
| Abused but not identified to dead | 0.0055 | 0.0029 | 0.0091 | Dirichlet | 14 |
| Stay in abused but not identified | 0.8762 | – | – | Dirichlet | Complement |
| Transition probabilities—control | | | | | |
| Abused but not identified to not abused | 0.0500 | 0.0412 | 0.0596 | Dirichlet | † |
| Abused but not identified to abused and identified, not seeing advocate | 0.0091 | 0.0055 | 0.0135 | Dirichlet | IRIS data |
| Abused but not identified to abused and identified, seeing advocate | 0.0226 | 0.0168 | 0.0293 | Dirichlet | IRIS data |
| Abused but not identified to dead | 0.0055 | 0.0029 | 0.0091 | Dirichlet | 14 |
| Stay in abused but not identified | 0.9131 | – | – | Dirichlet | Complement |

*Weighted average for adults and children.
†Internal calculation based on model calibration.
DVA, domestic violence/abuse; IRIS+, Identification and Referral to Improve Safety plus.

in the seven GP practices (99 337 patients) gives a six-month transition probability. For the usual care practices, this probability was estimated based on the number of women aged 16+ registered to GP practices in the same area referred to IRIS advocacy (39 382 patients).

3. Abuse not identified to abuse identified, not seeing advocate.

   We used the ratio of the number of patients abused and identified versus referred in the IRIS+ intervention to estimate the number of patients abused and identified, not seeing an advocate. These were effectively patients referred who decline support or who could not be contacted following the referral. The transition probability for usual care (IRIS intervention) was calculated as above, but only considered women identified vs referred.

4. Abuse not identified to no abuse.

   No data were available to reliably estimate this probability. We, therefore, estimated this using the model calibration method described below.

5. Abuse identified and seeing advocate to no abuse.

   This was taken from the MOSAIC (mothers' advocates in the community) trial,[27] identified in a Cochrane review,[28] evaluating the reduction of any type of domestic abuse with any type of advocacy.

6. Abuse identified, not seeing advocate to no abuse.

   This was taken from the control arm of the MOSAIC trial.[27]

7. No abuse to dead.

   We relied on the death rate per 1000 as estimated by the Office for National Statistics. For 2019, it was estimated at 10.4 per 1000. This implies the rate of dying per six months is 5.2 per 1000 people, excluding domestic homicides.

8. Abused to dead.

   For patients experiencing abuse this probability was 5.54 per 1000 (figure including domestic homicides) per six months. This estimate uses the Office for National Statistics death rate for 2019, including domestic homicides. For the purposes of the cost-effectiveness model, patients could not transition between the health states 'abuse identified, not seeing advocate' and 'abuse identified and seeing advocate'. This is because advocacy and support was offered to identified patients at point of referral and not reoffered. A patient could in principle self-refer into the support service later. However, if a patient self-referred after being identified by GP practice teams within six months, this would be considered a repeat referral and excluded from the model.

### Model calibration

We used the prevalence of abuse (5.5%) to calibrate the model, since there was uncertainty surrounding transition probabilities for 'no abuse to abuse not identified' and vice versa. The calibration was run for 3000 cycles, assuming that after this, the number of patients in each state would remain constant. The transition probabilities for 'no abuse to abuse not identified' and vice versa were changed until the proportion of patients in the 'no abuse' state exactly reflected the observed prevalence (100–5.5=94.5%). The initial steady state calculation showed that the probabilities from 'no abuse to abuse not identified' and 'abuse not identified to no abuse' needed adjusting. We then re-ran the calibration process using a prevalence of abuse figure of 17%, estimated in Richardson et al's study.[29] This led to an increase in the probability of 'abuse not identified to no abuse' from 0.005 to 0.033, which is in line with the finding that the prevalence of abuse identified at GP is higher than in the general population.[30] We assessed whether this increase significantly changed the results from the model in a univariate sensitivity analysis and concluded that it did not change the results significantly, although it contributed to its uncertainty. To compensate for this increase, we increase the probability of 'no abuse to abuse not identified' from 0.0027 to 0.0033. These adjustments meant that the model better reflected the population prevalence of abuse. The initial distribution of patients in the relevant states were 94.5% in 'no abuse', 5.3% in 'abuse not identified', 0.018% in 'abuse identified and seeing advocate' and 0.18% in 'abuse identified, not seeing advocate'.

### Utilities

Each state in the Markov model was associated with a utility score (table 2), allowing us to measure QALYs associated with IRIS+ and usual care (IRIS) based on the proportion of patients in each health state in each of the 20 cycles in the model. Utility scores were separately collected and calculated for men, women and children. For the health state 'no abuse' the utility was assumed to be 0.85 for adults and 0.95 for children, following published population norms.[31] A subset of adults and children identified from the IRIS+ intervention filled in a 12-Item Short Form Health Survey (SF-12) and Child Health Utility (CHU-9D) form, respectively. If support/advocacy was accepted, questionnaire data were requested at: (1) baseline, defined as when support/advocacy started and (2) between 6 and 10 months follow-up, defined as the period when support/advocacy ended. A validated mapping algorithm was used to transform SF-12 scores to Short-Form Six-Dimension (SF-6D) utilities.[32] The published SF-6D utilities were derived from a representative sample (n=611) of the UK adult population using the standard gamble valuation method. Similarly, a published value set was used to transform CHU-9D scores into utilities.[33] The published CHU-9D value set was derived from members of the UK adult population (n=300) using both standard gamble and ranking valuation methods. Estimated scores at baseline were attributed to 'abuse identified, not seeing advocate'. Follow-up scores were attributed to 'abuse identified and seeing advocate'. Due to the small number of forms collected (n=30 at baseline; n=16 at follow-up), these data were compared with previous literature for women for sense checking.[34] For 'abuse not identified',

**Table 2** Model input parameters: utilities and costs

| Parameter | Base case value | Lower limit | Upper limit | Distribution | Source |
|---|---|---|---|---|---|
| Utilities | | | | | |
| Not abused (adults) | 0.850 | 0.840 | 0.860 | Beta | 31 |
| Not abused (children) | 0.950 | 0.940 | 0.959 | Beta | 31 |
| Abused but not identified (women) | 0.656 | 0.522 | 0.749 | Beta | Assumption |
| Abused but not identified (men) | 0.626 | 0.500 | 0.744 | Beta | Assumption |
| Abused but not identified (children) | 0.801 | 0.623 | 0.932 | Beta | Assumption |
| Abused and identified, seeing advocate (women) | 0.659 | 0.518 | 0.782 | Beta | IRIS+ data |
| Abused and identified, seeing advocate (men) | 0.701 | 0.555 | 0.828 | Beta | IRIS+ data |
| Abused and identified, seeing advocate (children) | 0.804 | 0.625 | 0.935 | Beta | IRIS+ data |
| Abused and identified, not seeing advocate (women) | 0.656 | 0.522 | 0.749 | Beta | IRIS+ data |
| Abused and identified, not seeing advocate (men) | 0.626 | 0.500 | 0.744 | Beta | IRIS+ data |
| Abused and identified, not seeing advocate (children) | 0.801 | 0.623 | 0.932 | Beta | IRIS+ data |
| Costs (2019/2020£) | | | | | |
| Costs of the intervention, per patient exposed to DV, per 6 months | £0.75 | £0.02 | £2.73 | Gamma | IRIS+ budget |
| Cost of onward referral, once | £658 | £11 | £1908 | Gamma | IRIS+ data and IRIS data |
| Cost of Abused but not identified (weighted average–exposed population) | £4276 | £108 | £15 774 | Gamma | Weighted average* |
| Cost of abused but not identified (adults) | £4858 | £123 | £17 919 | Gamma | 35 |
| Cost of abused but not identified (children) | £1950 | £1000 | £2500 | Gamma | 20 |
| Weighted costs abused and identified, seeing advocate | 1 | 0.75 | 1.25 | Gamma | Assumption |
| Weighted costs abused and identified, not seeing advocate | 1 | 0.9 | 1.1 | Gamma | Assumption |

Costs are in 2019/2020 UK£.
*Excludes the cost of harms, which in this modelled are measured as benefits.
DV, domestic violence.

we assumed the utility score was the same as 'abuse identified, not seeing advocate', based on the assumption that identification alone (without advocacy support) does not improve quality of life.

### Costs

We included intervention costs, costs of onward referral and costs associated with abuse (including costs to the UK NHS, costs of lost economic output, costs to the criminal/civil justice system, personal costs) (table 2). Intervention costs were taken from the budget of the programme. The total budget for the delivery of IRIS+ was £60 253 and included salaries of the advocate educator and children worker, travel and consumables. This was divided by the total patient population exposed to the intervention (79 485 patients). The cost of onward referral considered the time an advocate educator or a children worker may spend working with external agencies (on average 57 hours), where their support alone would not suffice, multiplied by their average hourly salary (£29.60), and by 39%, which was the proportion of patients referred to the advocate or children's worker who accepted support and needed to be referred to another agency (57×£29.60×0.39=£658). The cost of onward referral

included the cost of referring men to the perpetrator programme. IRIS+ identified five men perpetrators, of which three engaged with the advocate educator. Of these, two accepted an onward referral to a perpetrator programme after risk assessment.

Costs associated with domestic violence in the UK for people aged 16+ is described in Oliver et al.[35] In this report, costs of lost economic output, health services, criminal justice system, civil justice system, social welfare, personal costs, specialised services and physical/emotional harm were included and unit cost per victim per year is estimated at £34 015 (in 2019 prices). We excluded costs of physical/emotional harm (£24 300), because in its report, Oliver et al calculate cost of physical/emotional harm by monetising QALY detriments. Since QALY gains are estimated for the intervention, including monetised QALY detriments in our costs was deemed inappropriate. This, however, implies that our results are conservative. For adults, the cost of abuse per six months was £4858. For children, we relied on a report produced by Pro Bono Economics,[20] which estimated the cost of domestic violence per child to be £1950 per six months in 2018£. We inflated this estimate (£1969 in 2019£). We considered

children to account for 20% of the UK population and estimated an overall cost of abuse per victim of £4276 (£4858×0.8–adults+£1969×0.2–children) per six months.

## Cost–utility analysis

A cost–utility analysis was conducted comparing costs and QALYs for IRIS+ versus IRIS (usual care). QALYs were calculated from utilities by using the area under the curve approach. The main outcome was the net monetary benefit (NMB) that estimates both costs and QALYs in monetary terms, using an acceptability threshold of £20 000 per QALY. A positive incremental NMB result indicates that IRIS+ intervention would be preferred on cost-effectiveness grounds. While a negative incremental NMB result indicates that the IRIS intervention (usual care) would be preferred. Results were also shown in terms of the incremental costs per QALY gained for IRIS+ versus IRIS. This was measured as the mean difference in costs between IRIS+ and IRIS divided by the mean difference in QALYs. We followed the usual decision-making rule for cost-effectiveness in the UK, in which an intervention is likely to be considered cost-effective when the incremental costs per QALY gained are less than £20 000.[19]

## Subgroups and distributional effects

The IRIS+ and IRIS arms represented two key groups which could be targeted in primary care (women, men and their children vs women only). Consequently, we did not estimate cost-effectiveness for any alternative subgroups. DVA is experienced across all social groups including all different socioeconomic, ethnicity and geographical groups. The IRIS and IRIS+ interventions are designed for all social groups, therefore, we did not consider distributional effects.

## Sensitivity analysis

We undertook a probabilistic sensitivity analysis, based on 1000 simulations drawn from random samples from the probability distributions of all parameters. These 1000 simulations were plotted in a cost-effectiveness plane. The proportion of simulations with an incremental cost per QALY gained below the cost-effectiveness threshold was calculated for different threshold values, ranging from £0 to £50 000. The results were presented in a cost-effectiveness acceptability curve.

## Patient and public involvement

Three patient and public involvement (PPI) groups (female survivors, male survivors and male perpetrators) were created and consulted throughout the lifetime of the research programme. PPI representatives were involved in the development of the IRIS+ intervention and the design of the research study.

## RESULTS

### Base case

The results of the cost–utility analysis in the base case analysis are presented in table 3. Over the 10-year time horizon, mean total costs per patient registered at GPs eligible to the IRIS+ intervention were £3867. For the IRIS intervention (usual care), the mean cost per patient was £3959. IRIS+, therefore, could potentially save £92 per patient over a 10-year time horizon. While a small sample may have contributed to the uncommon finding that the mean total costs in the intervention arm are smaller than the usual care, we identified that this difference is mainly a result of the number of patients that ultimately transition from 'abuse identified and seeing advocate to no abuse' and 'abuse identified, not seeing advocate to no abuse'. Given the IRIS+ intervention identifies (and supports) a larger proportion of patients than the control (see table 1), in our hypothetical cohort of 10 000, at the end of 20 six-monthly cycles, there are 8569 people in the 'not abused' health state in the intervention (IRIS+) arm and only 8538 in the usual care arm, thus preventing some cost of abuse in the IRIS+ intervention arm.

| Table 3 | Discounted base case and probabilistic results | | |
|---|---|---|---|
| **Discounted base case results** | **Costs** | **QALYs** | **Cost-effectiveness** |
| Intervention (IRIS+ programme) | £3867 | 7.000 | |
| Control (IRIS programme) | £3959 | 6.997 | |
| Difference (intervention vs control) | £-92 | 0.003 | Intervention dominates control |
| Incremental NMB* | | | £145 |
| **Probabilistic results** | **Costs (95% CI)** | **QALYs (95% CI)** | **Cost-effectiveness (95% CI)** |
| Intervention (IRIS+ programme) | £107 to £16 616 | 6.377 to 7.192 | |
| Control (IRIS programme) | £104 to £17 343 | 6.377 to 7.197 | |
| Increment | £−1123 to £171 | −0.030 to 0.019 | |
| ICER | | | £−206 828 to £277 989 |

Costs are in 2019/2020 UK£. Numbers may not sum due to rounding.
*Measured at a willingness to pay for a QALY of £20 000.
ICER, incremental cost-effectiveness ratio; IRIS+, Identification and Referral to Improve Safety plus; NMB, net monetary benefit; QALY, quality-adjusted life-year.

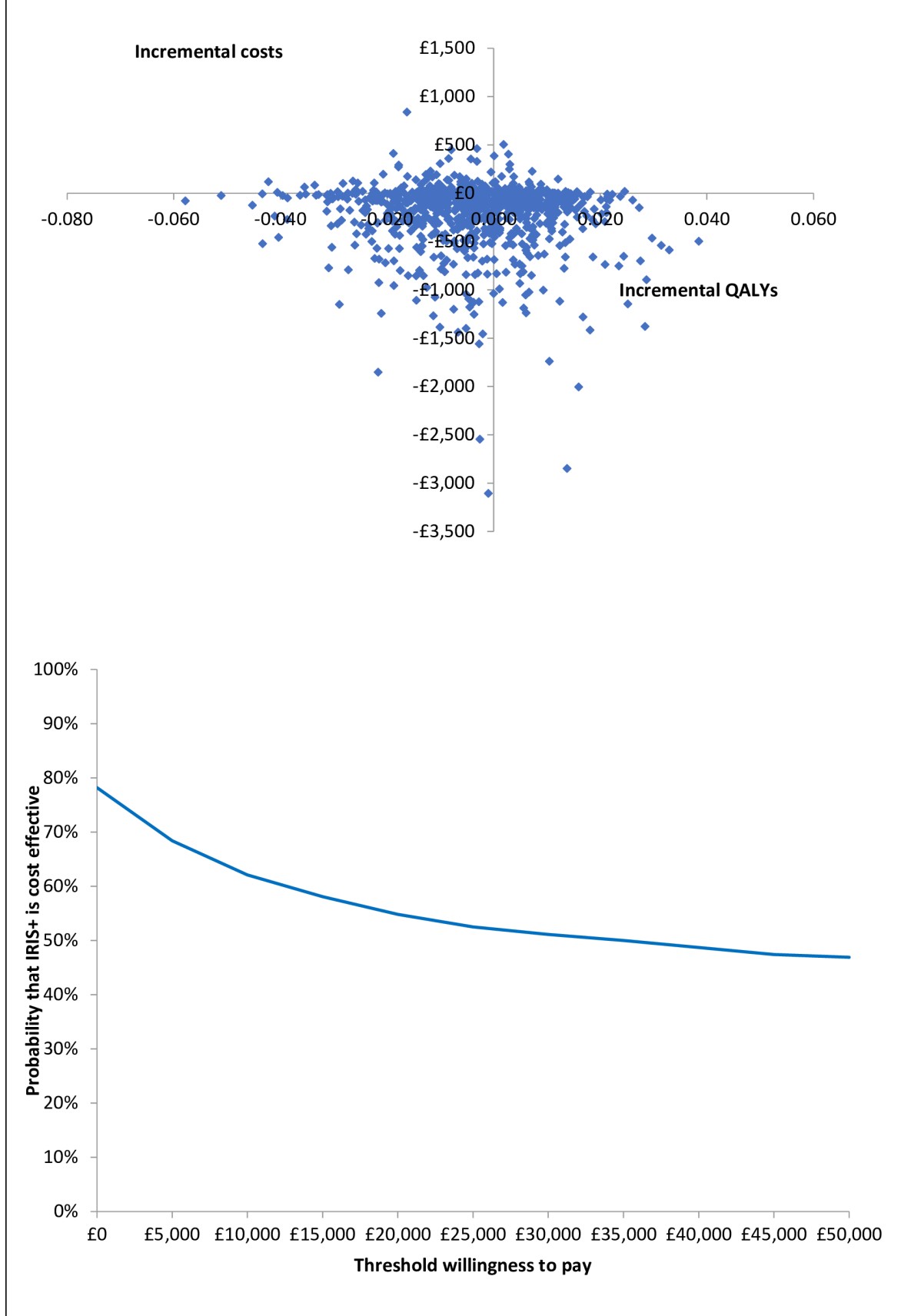

**Figure 2** Probabilistic sensitivity analysis. (A) Scatter plot of incremental costs and incremental QALYs from 1000 simulations. (B) Cost-effectiveness acceptability curve showing the probability that the intervention (IRIS+) is cost-effective versus control (IRIS) at different values of the maximum willingness to pay for a QALY. Costs are in 2019/2020 UK£. IRIS+, Identification and Referral to Improve Safety plus; QALY, quality-adjusted life year.

Total QALYs per patient were also 0.003 higher in the IRIS+ arm (7.000) than in the IRIS arm (usual care) (6.997). As the IRIS+ intervention arm was associated with lower costs and higher effectiveness then the incremental cost per QALY gained was negative (dominating usual care, IRIS) and the incremental NMB was positive (£145).

## Sensitivity analysis
Incremental costs and QALYs varied widely in the probabilistic sensitivity analyses. The 95% CI for incremental costs was –£1123 to £171, while for incremental QALYs it was –0.030 to 0.019, and the ICER was –£206 828 to £277 989 per QALY gained. Figure 2A shows a scatter plot of the incremental costs and incremental QALYs from the 1000 simulations. It shows how much uncertainty there is around these results. The IRIS+ intervention was cost-effective in 55% of simulations when the cost-effectiveness threshold was £20 000 (figure 2B).

## DISCUSSION
We found that the IRIS+ intervention could be cost-effective or even cost saving from a societal perspective in the UK with a willingness-to-pay threshold of £20 000 per gain in QALY, when compared with usual care (IRIS). There is considerable uncertainty surrounding these results, and only slightly more than a 50% probability that IRIS+ is cost-effective at £20 000 per QALY, the cost-effectiveness threshold commonly used in the UK.

There are a number of strengths and limitations to this study. The main strength relates to this study drawing on newly collected data, reducing the need for using out-of-date previously published estimates. It, however, relies on a small number of observations (n=30 at baseline; n=16 at follow-up), which could potentially be unreliable. The large uncertainty in our results reflects the small sample size. Nevertheless, as far as the authors are aware, this is the first study to assess the potential cost-effectiveness of a primary care intervention providing support to not just women, but also men and children experiencing DVA. Another important limitation of this study relates to its prospective nature. Given the pilot design, we were unable to directly recruit practices into IRIS+ and IRIS (usual care). Thus, by using practices in the same area, spillover effects may be significant (although they were not explored in this paper). Furthermore, the small number of practices, and as a result the small number of patients identified, meant that subgroup analysis was not possible. A cluster randomised controlled trial comparing IRIS+ to IRIS (usual care) could potentially address some of the uncertainties observed in the cost-effectiveness result of this study. More specifically, an economic evaluation conducted alongside a trial may shed light on some of the differences in terms of costs and benefits for women, men and children.

Comparing this prospective study to similar studies in the literature is challenging. Most training and advocacy programmes evaluated to date, have focused on a subset of the population, such as women, children or men only. Including all groups is a key strength of the IRIS+ intervention, as reported in the qualitative findings of this research study.[36] Future research should attempt to replicate the intervention in a greater number of GPs across the UK to enable more robust data collection and larger sample sizes.

**Contributors** Cochrane attests that all listed authors meet authorship criteria and that no others meeting the criteria have been omitted from the opportunity to be listed as an author. MC had primary responsibility for the overall content as the guarantor. MC, ES, CC, EE, MJ, GF and ECB contributed to the planning of the study. ES and CC managed the coordination of the study. MC and ECB conducted the analysis for the study and MC, ECB, ES, EE and GF contributed to the interpretation of the data. MC and ECB developed the manuscript. MC, ES, CC, EE, MJ, GF and ECB read and commented on the manuscript drafts and approved the final manuscript. GF was the chief investigator of the study. The study was funded by National Institute for Health Research (Programme Grants for Applied Research), (RP-PG-0614-20012).

**Funding** IRIS+ is part of the REPROVIDE programme (Reaching Everyone Programme of Research On Violence in diverse Domestic Environments), an independent research programme funded by the National Institute for Health Research (Programme Grants for Applied Research), (RP-PG-0614-20012).

**Disclaimer** The views expressed in this publication are those of the authors and not necessarily those of the National Institute for Health Research or the Department of Health and Social Care. The funder had no role in the design and conduct of the study; collection, management, analysis, and interpretation of the data; preparation, review, or approval of the manuscript; and decision to submit the manuscript for publication.

**Competing interests** MJ is the CEO of IRISi and was a named partner in REPROVIDE. She did not influence the economic modelling or its results.

**Patient and public involvement** Patients and/or the public were involved in the design, or conduct, or reporting, or dissemination plans of this research. Refer to the Methods section for further details.

**Patient consent for publication** Not applicable.

**Ethics approval** This study involves human participants and was approved by ethics committee reference: 19/LO/1132. The study was given favourable ethical approval by London—Hampstead Research Ethics Committee (REC reference: 19/LO/1132) and the Health Research Authority (HRA) and Health and Care Research Wales (HCRW). Participants gave informed consent to participate in the study before taking part.

**Provenance and peer review** Not commissioned; externally peer reviewed.

**Data availability statement** All data relevant to the study are included in the article or uploaded as online supplemental information.

**ORCID iDs**
Madeleine Cochrane http://orcid.org/0000-0003-1856-3293

Caroline Coope http://orcid.org/0000-0001-7803-8760
Gene Feder http://orcid.org/0000-0002-7890-3926

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
