## [Reviewer comments · BMJ Open]

ARTICLE DETAILS

TITLE (PROVISIONAL)	Primary care system-level training and support programme for the secondary prevention of domestic violence and abuse: a cost-effectiveness feasibility model
AUTHORS	Cochrane, Madeleine; Szilassy, Eszter; Coope, Caroline; Emsley, Elizabeth; Johnson, Medina; Feder, Gene; Barbosa, Estela

VERSION 1 – REVIEW

REVIEWER	Couden Hernandez, Barbara Loma Linda University, Office of Physician Vitality
REVIEW RETURNED	28-Feb-2023

GENERAL COMMENTS	This is a carefully presented paper. The inclusion of perpetrators in the cost analysis is somewhat problematic, in that recidivism rates for DVA perpetrators is typically quite high depending on the type of abuser they are. Nguyen & Bird, (Public Policy of California, 2018, https://www.ppic.org/blog/tailoring-domestic-violence-programs-to-reduce-recidivism/) report a 67% rearrest rate within two years. Hence, the services required for perpetrators most likely include individual, group, educational and legal oversight in other countries. Indeed, the costs of perpetrator treatment might be quite different from perpetrator victim services. If the authors have factored this fact into their model it should be acknowledged to ensure reliability. If not, please clarify why perpetrators are included in costs cited. A minor point is the placement of "Dead" in Figure 1. As one reads from left to right, it is counterintuitive to list death on the left of the figure, as if preceding the other state categories on the model. A few misspellings should be corrected (aonger, anti-contgion measures, challenging, etc.).
--

REVIEWER	Weber, Ellerie Icahn School of Medicine at Mount Sinai
REVIEW RETURNED	07-Apr-2023

GENERAL COMMENTS	Comments: This study is a cost-utility analysis from the societal perspective that compares an intervention (IRIS+) to a control (IRIS). IRIS+ identifies, documents and refers (to advocacy) women, men & children who may have experience domestic violence/abuse. Comparatively, IRIS is a similar program that only focuses on women. The authors find that IRIS+ saves \$93GBP/patient over 10yrs and has QALY gains of 0.003, but that it is inly cost-effective in about half of simulations when using a CE threshold of \$20,000GBP.
--

This is a well-written paper that adds to a small literature on the cost-effectiveness of DV prevention interventions, and thus is a contribution. There are a few methodological items that the authors should clarify before I would recommend for publication.

Abstract

(1) Authors talk about IRIS+ identifying individuals who may be perpetrators, but paper is entirely focused on the perspective of those abused – needs to be reconciled.

Background/Overview

(1) How many control practices (ie sites in IRIS) are there in addition to 7 GP practices (ie sites in IRIS+), or were they at the same sites? Was this not an RCT? The authors mention this in the discussion. A little more description of the intervention would be warranted (I know the authors cite other pubs describing the intervention, but more detail should be in this manuscript).

Model structure

- (1) What software did authors use to run Markov model?
- (2) Explain why 6 month cycle length was chosen

Transition Probabilities

(1) P 12- Line 49-56 – Denominator is # of eligible patients, what does “eligible patients” mean? Really want denominator to be the # that are abused, correct? Please clarify

(2) P 13 –Line 3-6 – “For the control practices, this probability was estimated based on the number of women only referred to IRIS advocacy.”

a. What’s the denominator for the control practices?

(3) P 13- Line 10-12 – “We used the ratio of the number of patients abused and identified vs referred in the IRIS+ pilot to estimate the number of patients abused and identified”. Typo? Authors mean to estimate the number to patients abused and *not* identified?

(4) P 13- Line 52-56 – Please clarify where authors get this 5.54/1000 number from? From IRIS?

(5) Why is it impossible in model to move from “Abused and identified with no advocate” to “Abused and Identified with advocate”?

(6) It would be helpful for the authors to provide more information about how the model calibration was executed. The initial distribution of the 3 abused categories had to equal 5.5%? And then how did they calibrate the transition probabilities?

Utilities

(1) P 15 – Line 13 – Typo? Should be “... assumption that identification alone (with*out* advocacy support)”

Costs

(1) P 15 – Line 24-26 – Authors should explain more about the intervention costs and where this program budget number came from. For example: Did they just have a total budget (that they then divided by the # of participants)? Was the budget calculated by the researchers from invoices? Does it include materials and labor, or what exactly was included?

(2) P 15 – Line 26-31 – Similar to previous point, more about the onward referral cost calculation is required. For example: Was this taken from intervention data that captured the # of hours advocated worked with external agencies? Multiplied by hourly wages? Wages adjusted for anything? etc

(3) P 15 – Categories included in costs – This is my biggest concern in the paper. I don’t understand how the authors justify excluding the costs of physical/emotional harm (ie, including them

	as a benefit of averting abuse, ie, harm averted), but not excluding the health care costs and social costs (criminal justice, civil justice, etc). Don't the social and health care costs also stem from "harms" that would be averted if DV/abuse was prevented and thus should be considered benefits from the intervention? Can the authors explain this choice and how they justify it? a. I would have expected costs of the intervention to be higher (e.g., in IRIS+ there is a dedicated children's worker), but the benefits to be higher (presumably bc they are averting more abuse/violence in children's and men's lives. (4) P 15 – Line 47-52 – I don't understand the adjustment of the cost of DV per child from \$1950GBP to \$4276 GBP. Some of that I believe is from inflation from 2018 to 2019? However, why is the % of the population that is children relevant since this is a per child cost? Please clarify. Modelling choices – subgroup and distributional effects (1) I understand that the sample size is very small, and likely prohibits doing subgroup analysis, but it would be good to look at the CE just among women, children and men. If benefits/effectiveness are found among women, why would IRIS+ be more effective than IRIS? It would strengthen the authors argument if they were able to find CE among children/men.
--	--

VERSION 1 – AUTHOR RESPONSE

Item	Reviewer 1's comments:	Authors response to explain any changes that have (or have not) been made to the original article, being as specific as possible.
General comment	This is a carefully presented paper.	No action required, general comment
1	The inclusion of perpetrators in the cost analysis is somewhat problematic, in that recidivism rates for DVA perpetrators is typically quite high depending on the type of abuser they are.  • Nguyen & Bird, (Public Policy of California, 2018, https://www.ppic.org/blog/tailoring-domestic-violence-programs-to-reduce-recidivism/) report a 67% rearrest rate within two years. • Hence, the services required for perpetrators most likely include individual, group, educational and legal oversight in other countries. • Indeed, the costs of perpetrator treatment might be quite different from perpetrator victim services. • If the authors have factored this fact into their model it should be acknowledged to ensure reliability. 	Thank you for the helpful literature on reincidence of abuse by type of abuser. The perpetrator group programme was linked to the IRIS+ intervention via a referral pathway or signposting, but was not part of the intervention per se. Perpetrators could also self-refer into the perpetrator program. We have clarified this in the description of the IRIS+ intervention (abstract, pg.4, IRIS+ intervention, pg. 9).

	 If not, please clarify why perpetrators are included in costs cited. 	The cost of onward referral included the cost of referring men to the perpetrator programme. IRIS+ identified five men perpetrators, of which three engaged with the advocate educator. Of these, two accepted an onward referral to a perpetrator programme after risk assessment (methods, pg.7, costs, pg.17).
2	A minor point is the placement of "Dead" in Figure 1. As one reads from left to right, it is counterintuitive to list death on the left of the figure, as if preceding the other state categories on the model.	We have changed the position of Dead in Figure 1. New figure uploaded.
3	A few misspellings should be corrected (aonger, anti-contgion measures, challenging, etc.).	Thank you we have proofread the document and corrected these spelling errors.

Item	Reviewer 2's comments: Dr. Ellerie Weber, Icahn School of Medicine at Mount Sinai	Authors response to explain any changes that have (or have not) been made to the original article, being as specific as possible.
General comment	This study is a cost-utility analysis from the societal perspective that compares an intervention (IRIS+) to a control (IRIS). IRIS+ identifies, documents and refers (to advocacy) women, men & children who may have experience domestic violence/abuse. Comparatively, IRIS is a similar program that only focuses on women. The authors find that IRIS+ saves \$93GBP/patient over 10yrs and has QALY gains of 0.003, but that it is inly cost-effective in about half of simulations when using a CE threshold of \$20,000GBP.	No action required, general comment
General comment	This is a well-written paper that adds to a small literature on the cost-effectiveness of DV prevention interventions, and thus is a contribution. There are a few methodological items that the authors should clarify before I would recommend for publication.	No action required, general comment
1	Abstract	The perpetrator group programme was linked to the IRIS+ intervention via a referral pathway or signposting, but

	Authors talk about IRIS+ identifying individuals who may be perpetrators, but paper is entirely focused on the perspective of those abused – needs to be reconciled.	was not part of the intervention per se. Perpetrators could also self-refer into the perpetrator program. We have clarified this in the description of the IRIS+ intervention (abstract, pg.4, IRIS+ intervention, pg. 9).
2	Background/Overview How many control practices (ie sites in IRIS) are there in addition to 7 GP practices (ie sites in IRIS+), or were they at the same sites? Was this not an RCT? The authors mention this in the discussion. A little more description of the intervention would be warranted (I know the authors cite other pubs describing the intervention, but more detail should be in this manuscript)	As this was a feasibility study and not a RCT, we used data from the same sites. We have added information to explain that 4 of the 7 practices recruited into IRIS+ had previously received IRIS training, while other 3 were completely naïve to IRIS+ or IRIS training (comparisons between IRIS+ and IRIS, pg. 9). We also added that future research should conduct a cluster RCT, including a trial economic evaluation (discussion, pgs. 24 and 25).
3	Model structure (1) What software did authors use to run Markov model? (2) Explain why 6 month cycle length was chosen	(1) We have added information that the model was built and run in Excel VBA (model structure, pg.8) (2) The 6 month cycle is justified to reflect the average length of support received from advocacy services following referral and the fact that 6 months is the maximum time of support available for identified patients (model structure, pg.8).
4	Transition Probabilities P 12- Line 49-56 – Denominator is # of eligible patients, what does “eligible patients” mean? Really want denominator to be the # that are abused, correct? Please clarify	We considered using the abused population as a denominator, but that is not observable, so would be an estimate (based on the prevalence of abuse). Since the interventions are both designed to identify abuse, we considered the denominator the number of patients registered to GP practices, since they would all be exposed to the intervention(s). For IRIS+, the denominator is the total patient population registered at the 7 GP practices (99337) and for IRIS, the

		denominator was the number of women aged 16+ registered to GP practices registered in the same area (39382). We clarified this in the text (Abuse no identified to Abuse identified and seeing advocate, pgs. 13 and 14).
5	Transition Probabilities P 13 –Line 3-6 – “For the control practices, this probability was estimated based on the number of women only referred to IRIS advocacy.” a. What’s the denominator for the control practices?	As mentioned above, for consistency, we used the patient population exposed to the intervention. For IRIS, the denominator was the number of women aged 16+ registered to GP practices registered in the same area (39382). We clarified this in the text (Abuse no identified to Abuse identified and seeing advocate, pgs. 13 and 14).
6	Transition Probabilities P 13- Line 10-12 – “We used the ratio of the number of patients abused and identified vs referred in the IRIS+ pilot to estimate the number of patients abused and identified”. Typo? Authors mean to estimate the number to patients abused and *not* identified?	Thank you. We corrected the typo (Abuse not identified to Abuse identified, not seeing advocate, pg.14)
7	Transition Probabilities P 13- Line 52-56 – Please clarify where authors get this 5.54/1000 number from? From IRIS?	The 5.54/1000 probability of dying is based on the Office for National Statistics death rate for 2019, including domestic homicides. We have clarified in the text (Abused to Dead, pg.14)
8	Transition Probabilities Why is it impossible in model to move from “Abused and identified with no advocate” to “Abused and Identified with advocate”?	Advocacy and support is offered at point of referral and not re-offered. A patient can later self-refer. If a patient self-referred after being identified by GP practice teams within 6 months, this would be considered a repeat referral (and excluded from the model). We have clarified this in the text (transition probabilities, pg. 15).
9	Transition Probabilities	We have added detail to how the model was calibrated. The initial distribution of

	It would be helpful for the authors to provide more information about how the model calibration was executed. The initial distribution of the 3 abused categories had to equal 5.5%? And then how did they calibrate the transition probabilities?	all abuse categories had to be equal to 5.5% (prevalence of abuse). We needed to adjust the probabilities of 'Abuse not identified to No abuse' and No abuse to Abuse not identified' to ensure the model reflected the correct prevalence of abuse in the population (model calibration, pg.15)
10	Utilities P 15 – Line 13 – Typo? Should be "... assumption that identification alone (with*out* advocacy support)"	Thank you. Typo corrected (Utilities, pg. 16)
11	Costs P 15 – Line 24-26 – Authors should explain more about the intervention costs and where this program budget number came from. For example: Did they just have a total budget (that they then divided by the # of participants)? Was the budget calculated by the researchers from invoices? Does it include materials and labor, or what exactly was included?	We have added the total cost of the intervention and what it includes. We also added detail on how the cost per patient was calculated and realised we had a small typo in the cost (£0.74 was corrected to £0.75) (costs, Table 2 on pg. 19 and Supplementary Material Tables 1 and 2).
12	P 15 – Line 26-31 – Similar to previous point, more about the onward referral cost calculation is required. For example: Was this taken from intervention data that captured the # of hours advocated worked with external agencies? Multiplied by hourly wages? Wages adjusted for anything? Etc	We have added an explanation of how cost of onward referral was calculated (average number of hours of support x hourly wage x proportion of patients needing onward referral) (costs, pg. 17).
13	P 15 – Categories included in costs – This is my biggest concern in the paper. I don't understand how the authors justify excluding the costs of physical/emotional harm (ie, including them as a benefit of averting abuse, ie, harm averted), but not excluding the health care costs and social costs (criminal justice, civil justice, etc). Don't the social and health care costs also stem from "harms" that would be averted if DV/abuse was prevented and thus should be	In the most up-to-date estimate of cost of abuse in the UK, the cost of physical / emotional harms is calculated by multiplying £60,000 (value of QALY per year) by QALY detriments to physical and mental health. Since our model looks at the health-related quality of life impact in the estimation of QALY gains (benefits), it felt inappropriate to include the monetised QALY detriments in the costs. If anything, our model is a conservative estimate of prospective cost-effectiveness of the IRIS+

	considered benefits from the intervention? Can the authors explain this choice and how they justify it? a. I would have expected costs of the intervention to be higher (e.g., in IRIS+ there is a dedicated children’s worker), but the benefits to be higher (presumably bc they are averting more abuse/violence in children’s and men’s lives.	intervention. We have added a brief explanation to the main manuscript (costs, pg. 17).
14	P 15 – Line 47-52 – I don’t understand the adjustment of the cost of DV per child from \$1950GBP to \$4276 GBP. Some of that I believe is from inflation from 2018 to 2019? However, why is the % of the population that is children relevant since this is a per child cost? Please clarify.	We apologise for the confusing phrasing. In fact the estimated cost per patient is the weighted average between children (20%) and adults (80%). We have changed the sentence to improve clarity (costs, pg. 17-18).

VERSION 2 – REVIEW

REVIEWER	Weber, Ellerie Icahn School of Medicine at Mount Sinai
REVIEW RETURNED	18-Sep-2023

GENERAL COMMENTS	I thank the authors for their thorough response and think the paper has improved as a result. I still have some comments that should be addressed before publication. Major comments: I have some concerns about the calibration methods used and resulting transition probabilities. (1) On p15, the authors write: “We assumed some patient would no longer be exposed to abuse naturally and increased the probability of ‘Abuse not identified to No abuse’ from 0.005 to 0.033.” This is a big change (more than 6-fold!); what calibration method exactly did the authors use to make this determination? Was there an assessment of fit, or any sensitivity around it? More explanation would be helpful. (2) Table 1 is confusing: a. Transition probabilities have one row repeated (“Abused and identified, not seeing advocate to Dead”)- typo? b. 8 transition probability groups discussed in text, but 19 lines (18 if one repeat was a typo) in Table. Even if each 8 had a different # for the intervention & control groups, that should only make 16? Please explain/fix. In general, there may be better ways to present this information to make less confusing for the reader. E.g., have a figure showing the probabilities, break the table up into (or have different columns for) transition probabilities for intervention vs those for control, etc c. Patients experiencing abuse – currently sub-lists prevalence of DVA in adults, then in children. Confusing both
---

	because they add up (.055+.08>.055) to more than the total pop probability – ie, doesn't tell reader that it's a weighted sum. Also omits the 65+ category listed in text. Please do the following  i. Change "Proportion of patients experiencing abuse" to "Proportion of patients experiencing abuse – all ages" ii. Add sub-line for 65+ pop iii. Consider putting the proportion weights of each sub-line population into the table I also still have some concern about the costs/conclusions (3) The authors did not address my previous comment "I would have expected costs of the intervention to be higher (e.g., in IRIS+ there is a dedicated children's worker), but the benefits to be higher (presumably bc they are averting more abuse/violence in children's and men's lives). Meaning, they find costs higher in the control group (£3,959) than the intervention (£3,867), but don't really discuss this oddity explicitly in the paper. Why would this occur? Are they saying this is only a result of a small n?" (4) It is hard to conclude the intervention dominates the control, nor as the authors write "We found that the IRIS+ intervention is likely to be cost-effective or even cost-saving from a societal perspective in the UK with a willingness-to-pay threshold of £20,000 per gain in QALY, when compared to usual care (IRIS)." It only has a 50-50 chance of being CE!  a. I would recommend the authors walk back this conclusion in the paper and the abstract as well. I think the paper still has merit (e.g., there is such a lack of CE data on DV interventions) even without concluding that it is cost effective. Minor comment: (1) Abstract line should be edited from "A perpetrator group programme was linked to the IRIS+ intervention via a referral pathway or signposting, and not part of the intervention per se." to something more like "A perpetrator group programme was not part of the intervention per se, but was linked to the IRIS+ intervention via a referral pathway and signposting."
--	---

VERSION 2 – AUTHOR RESPONSE

Reviewers Comments	Response	Revised text
Reviewer 2		
I have some concerns about the calibration methods used and resulting transition probabilities. (1) On p15, the authors write: "We assumed some patient would no longer be exposed to abuse naturally and increased the probability of 'Abuse not identified to No abuse' from 0.005 to 0.033." This is a big change (more than 6-fold!); what calibration	We apologise for not originally including more details on how the model calibration was conducted. We have now added more detail, including in relation to an univariate sensitivity analysis of the parameter 'Abuse not identified to No abuse'.	The initial steady state calculation showed that that the probabilities from 'No abuse to Abuse not identified' and 'Abuse not identified to No abuse' needed adjusting. We then re-ran the calibration process using a prevalence of abuse figure of 17%, estimated in Richardson and colleagues' study(1). This led to an increase in

method exactly did the authors use to make this determination? Was there an assessment of fit, or any sensitivity around it? More explanation would be helpful.		the probability of 'Abuse not identified to No abuse' from 0.005 to 0.033, which is in line with the finding that prevalence of abuse identified at general practice is higher than in the general population (2). We assessed whether this increase significantly changed the results from the model in a univariate sensitivity analysis and concluded that it did not change the results significantly, although it contributed to its uncertainty.
(2) Table 1 is confusing: a. Transition probabilities have one row repeated ("Abused and identified, not seeing advocate to Dead")- typo?	Thank you, we have checked and we have included 'Abused and identified, not seeing advocate to Dead' only once. We have included a transition probability with a similar name but it refers to people who are seeing an advocate, this transition probability is called 'Abused and identified, seeing advocate to Dead'.	No change needed.
b. 8 transition probability groups discussed in text, but 19 lines (18 if one repeat was a typo) in Table. Even if each 8 had a different # for the intervention & control groups, that should only make 16? Please explain/fix.	There are more than 16 probabilities because we have provided information about the proportion of people who do not transition per cycle. This is referred to as the complement.	No change needed.
In general, there may be better ways to present this information to make less confusing for the reader. E.g., have a figure showing the probabilities, break the table up into (or have different columns for) transition probabilities for intervention vs those for control, etc c. Patients experiencing abuse – currently sub-lists prevalence of DVA in adults, then in children. Confusing	Thank you for your suggestion we have presented the table slightly differently by adding in three additional rows as subheadings, to indicate transition probabilities relate to the intervention vs those for the control. We have amended the text of the table so as it reads 90+ years.	"Transition probabilities-intervention and control" "Transition probabilities-intervention" "Transition probabilities-control" "Proportion of people experiencing abuse- all ages" and "Proportion of

both because they add up (.055+.08>.055) to more than the total pop probability – ie, doesn't tell reader that it's a weighted sum. Also omits the 65+ category listed in text.	We have provided a footnote to explain this is a weighted average.	people experiencing abuse- aged 16 to 90+ years” ***Weighted average for adults and children”
Please do the following i. Change “Proportion of patients experiencing abuse” to “Proportion of patients experiencing abuse – all ages” ii. Add sub-line for 65+ pop iii. Consider putting the proportion weights of each sub-line population into the table	i) Thank you for your suggestion we have changed to your suggested wording ii) We have amended the text in the table to 90+ years iii) Our supplementary tables provide the breakdown by subgroup.	“Starting distribution for those experiencing abuse- all ages” “Prevalence of DVA in adults (males and females) – aged 16 to 90+”
I also still have some concern about the costs/conclusion (3) The authors did not address my previous comment “I would have expected costs of the intervention to be higher (e.g., in IRIS+ there is a dedicated children’s worker), but the benefits to be higher (presumably bc they are averting more abuse/violence in children’s and men’s lives). Meaning, they find costs higher in the control group (£3,959) than the intervention (£3,867), but don’t really discuss this oddity explicitly in the paper. Why would this occur? Are they saying this is only a result of a small n?	We apologise for the oversight in not addressing your comment at first round. While the small n may indeed contribute to this oddity, what mainly explains this difference is that fact that a larger proportion of patients identified are seen by the advocate in the intervention arm (0.0312) than in the control (0.0226). As a result, a larger number of patients transitions from ‘Abuse identified seeing advocate to Not Abuse’ in the intervention arm (0.0312 x 0.1408) than in the control arm (0.0226 x 0.1408). A similar thing happens to the proportion of patients identified not seeing an advocate. It is by increasing the number of patients in the ‘Not abused’ state that the cost of abuse is prevented. In our hypothetical cohort of 10,000, at the end of 20 6-monthly cycles, there are 8,569 people in the ‘Not Abused’ health state in the intervention (IRIS+) arm and only 8,538 in the usual care arm.	While a small sample may have contributed to the uncommon finding that the mean total costs in the intervention arm is smaller than the usual care, we identified that this difference is mainly a result of the number of patients that ultimately transition from ‘Abuse identified and seeing advocate to No abuse’ and ‘Abuse identified, not seeing advocate to No abuse’. Given the IRIS+ intervention identifies (and supports) a larger proportion of patients than the control (see Table 1), in our hypothetical cohort of 10,000, at the end of 20 6-monthly cycles, there are 8,569 people in the ‘Not Abused’ health state in the intervention (IRIS+) arm and only 8,538 in the usual care arm, thus preventing some cost of abuse in the IRIS+ intervention arm.

	We have included this explanation in the main manuscript.	
(4) It is hard to conclude the intervention dominates the control, nor as the authors write “We found that the IRIS+ intervention is likely to be cost-effective or even cost-saving from a societal perspective in the UK with a willingness-to-pay threshold of £20,000 per gain in QALY, when compared to usual care (IRIS).” It only has a 50-50 chance of being CE! a. I would recommend the authors walk back this conclusion in the paper and the abstract as well. I think the paper still has merit (e.g., there is such a lack of CE data on DV interventions) even without concluding that it is cost effective	Thank you for your comment. We agree that the results are less positive than originally outlined and have taken a more nuanced stance now. We made changes to the abstract and to the discussion.	Abstract: The IRIS+ intervention could be cost-effective or even cost-saving from a societal perspective in the UK, though there are large uncertainties, reflected in the confidence intervals and simulation results. Discussion: We found that the IRIS+ intervention could be cost-effective or even cost-saving from a societal perspective in the UK with a willingness-to-pay threshold of £20,000 per gain in QALY, when compared to usual care (IRIS). There is considerable uncertainty surrounding these results, and only slightly more than a 50% probability that IRIS+ is cost-effective at £20,000 per QALY, the cost-effectiveness threshold commonly used in the UK.
Minor comment: (1) Abstract line should be edited from “A perpetrator group programme was linked to the IRIS+ intervention via a referral pathway or signposting, and not part of the intervention per se.” to something more like “A perpetrator group programme was not part of the intervention per se, but was linked to the IRIS+ intervention via a referral pathway and signposting”	Thank you for your suggestion, we have made the suggested amendment in the abstract.	“A perpetrator group programme was not part of the intervention per se, but was linked to the IRIS+ intervention via a referral pathway and signposting”

References:

1. Richardson J, Coid J, Petruckevitch A, Chung WS, Moorey S, Feder G. Identifying domestic violence: cross sectional study in primary care. *Bmj*. 2002;324(7332):274.
2. Feder G, Ramsay J, Dunne D, Rose M, Arsene C, Norman R, et al. How far does screening women for domestic (partner) violence in different health-care settings meet criteria for a screening programme? Systematic reviews of nine UK National Screening Committee criteria: Chapter 4 What is the prevalence of partner violence against women and its impact on health? (Question I). *Health Technol Assess*. 2009;13(16):iii-iv, xi-xiii, 1-113, 37-347.